# Multichannel Sea Clutter Measurement and Space-Time Characteristics Analysis with L-Band Shore-Based Radar

**Jintong Wan [1,2], Feng Luo [1,*], Yushi Zhang [2], Jinpeng Zhang [2] and Xinyu Xu [2]**

[1] National Laboratory of Radar Signal Processing, Xidian University, Xi'an 710071, China
[2] National Key Laboratory of Electromagnetic Environment, China Research Institute of Radiowave Propagation, Qingdao 266107, China
[*] Correspondence: luofeng@xidian.edu.cn; Tel.: +86-029-8820-1031

**Abstract:** In order to study the space-time characteristics of sea clutter, the sea clutter is always measured by the airborne multichannel radar; however, the sea clutter shows the heterogeneity between range gates, which means the space-time covariance matrix's correspondence to the single range gate cannot be estimated accurately. Meanwhile, the measurement of the sea clutter data by the airborne radar is usually affected by the motion of the platform, which makes the analysis results unrepresentative of the space-time characteristics of the pure sea clutter. In this paper, a sea clutter measurement method based on L-band shore-based multichannel radar is proposed, where the transmit sub-array periodically moves with the pulse repetition period to obtain multiple sets of coherent processing interval pulses for each range gate. This measurement method can exclude the influences of the moving platform. Moreover, a sea clutter space-time signal model of the single range gate is proposed, and the model is used to simulate three-dimensional sea clutter data with space-time coupling characteristics. With verification of the measured and simulated data, it can be seen that the data composed of single range gate and multiple coherent processing interval pulses can accurately estimate the space-time covariance matrix corresponding to this single range gate. Furthermore, the space-time characteristics are analyzed based on the measured data. The results show that the eigenvalue spectrum and the spread width of space-time power spectrum are influenced by the backscattering coefficient of sea clutter and the speed of sea surface motion. In comparison, the decorrelation effect caused by the backscattering coefficient of sea clutter is stronger than that caused by the speed of the surface motion. The proposed method is helpful for guiding multichannel sea clutter measurement and the analysis results are of great significance to the clutter suppression algorithms of the marine multichannel radar.

**Keywords:** sea clutter; multichannel radar; space-time adaptive processing; measurement; space-time coupling characteristics

## 1. Introduction

The clutter problem has been accompanied by the development of airborne radar technology. The traditional one-dimensional time-domain filtering method has limited effect on airborne radar clutter suppression, which greatly limits the target detection performance of airborne radar. With the development of radar signal processing technology, the space-time adaptive processing (STAP) technology combines the space and time information of airborne multichannel radar to perform space-time two-dimensional filtering on clutter. It greatly improves the target detection performance of airborne multichannel radar for the ground clutter [1–5]. It is demonstrated in [6] that for certain scenarios, STAP can reliably detect small boats in the background of sea clutter. However, the evaluation of STAP suppression performance under the background of sea clutter is mostly at the level of theoretical simulation and modeling analysis, and there are few related studies based on the measured data [6,7].

To apply STAP to a maritime radar system, it is desirable to know in advance which theoretical STAP performance can be expected for that particular system, and the measured data is needed to verify and estimate this. Therefore, to evaluate the STAP performance, obtaining multichannel radar clutter measured data is very important. There are two ways to obtain sea clutter data measured by multichannel radar. The first method is to directly carry out the airborne multichannel radar sea clutter measurement test. In the 1990s, the United States implemented the Multichannel Airborne Radar Measurement (MCARM) program led by Rome Laboratory, which installed L-band radar composed of 24 channels on the aircraft and measured a large amount of clutter data reflecting the real environment of airborne radar [8]. In 2012, with the phased array multifunctional imaging radar, which operates at X-band, the FHR carried out a series of airborne multichannel clutter measurement experiments in Germany and collected real multichannel sea clutter data at different swell directions and different sea states [9–12]. McDonald established a multi-phase center coherent radar sea clutter model, and, based on PAMIR radar, measured data to verify the model [13,14]. The second method is to obtain measured sea clutter data with the shore-based multichannel radar. In 1993, the US Defense Advanced Research Projects Agency (DARPA) implemented the Mountaintop program, in which the sea clutter was measured by the shore-based radar using the technology of Inverse Displaced Phase Center Antenna (IDPCA) [15]. In reference [16], there exists a sea clutter measurement based on the technology of IDPCA and transceiver co-location technology, which can emulate platform motion and obtain sea clutter data with space-time coupling characteristics under various marine environmental parameters. However, there is one obvious feature in the above two measurement methods. The data measured by the above methods is composed of multiple range gates and one coherent processing interval (CPI) pulses, while the sea clutter amplitude varies with the range gates. Instantaneous sampling data cannot comprehensively reflect the space-time heterogeneity of sea clutter. Therefore, the space-time covariance matrix corresponding to the single range gate is only estimated by one CPI, which makes the analysis of the sea clutter space-time characteristics' inhomogeneity difficult.

Compared with land clutter, sea clutter is different due to the varying motion of the sea surface and due to different sea scattering types. Due to the wind, the speed and backscattering coefficient of sea clutter vary with the time and the sea state; the important implication of this difference are the broadening of the space-time filter notch and a bigger eigenvalue number of sea clutter [17–19]. Generally, the analysis of space-time inhomogeneity of sea clutter is mainly reflected in the following aspects. The first is the description of the motion characteristics of sea surface. The two-scale model [20] combines the Bragg model with the geometric model of sea surface with radar signals. Based on this model, Xin [17] incorporates the motion model of sea surface into the airborne sea clutter model. The second is the description of sea clutter in time and range dimensions. For the time dimension, there are many statistical models to provide a description of the distribution of sea clutter, such as the Rayleigh distribution [21], the Weibull distribution [22], the K distribution [21], the KK distribution [23], etc. The range dimension is mainly reflected in the backscattering coefficient of sea clutter. There are many empirical models in the literature, such as the Georgia Institute of Technology (GIT) model [21], the Technology Service Corporation (TSC) model [20], and the Naval Research Laboratory (NRL) model [24]. However, the above-mentioned inhomogeneity of sea clutter can only be reflected in theoretical analysis. At present, the inhomogeneity analysis of sea clutter based on measured data usually mixes the effect of backscattering coefficient of sea clutter and the speed of sea surface motion; it cannot be analyzed separately. Therefore, the influence of the backscattering coefficient of sea clutter and the speed of sea surface on the space-time characteristics cannot be analyzed accurately based on the measured data.

Motivated by the previous works, this paper proposes a method of the multichannel sea clutter measurement based on shore-based radar. In this paper, the measurement method that the transmit array periodically moves with the pulse repetition period is applied to obtain multiple CPIs for each range gate. A space-time signal model for single range

gate is proposed, which can show that the main influence on the space-time characteristics of sea clutter are the backscattering coefficient of sea clutter and the speed of sea surface. Moreover, this paper proposes a method for simulating three-dimensional sea clutter data which is based on the space-time signal model of the single range gate. In addition, the influence of the backscattering coefficient of sea clutter and speed of the sea surface on the eigenvalue spectrum and space-time power spectrum are analyzed, respectively, based on the measured data using the above multichannel sea clutter measurement system.

The reminder of this paper is organized as follows. In Section 2, multichannel sea clutter measurement is researched from three parts: the principle of measurement, the multichannel measurement system, and the processing method of the measured data. In Section 3, the measurement and space-time processing method are validated by comparing the processing results of the simulated and measured data. In Section 4, the space-time characteristics of sea clutter in different range gates are compared with the same sea state, also comparing that in different sea states with the same clutter to noise ratio (CNR), then analyzing the decorrelation effect caused by the scattering coefficient and the speed of sea surface, respectively. Finally, conclusions are drawn in Section 5.

## 2. Materials and Methods

The research on the multichannel sea clutter measurement solves three problems. The first question, involving shore-based multichannel radar, is how to generate sea clutter data with space-time coupling characteristics. The second question, involving multichannel radar measures, is in which way can data be generated that can be used to study the inhomogeneity of sea clutter. The third question is how to properly process the measured sea clutter data to reflect the space-time characteristics of pure sea clutter. This section answers each question one by one.

### 2.1. The Principle of Multichannel Sea Clutter Measurement System

Multichannel sea clutter measurement system is realized by the IDPCA technology; however, there is no clear explanation on why IDPCA can generate the clutter data with space-time coupling characteristics. In the following section the essence of space-time coupling characteristics is clearly introduced, then the realization of the space-time coupling characteristics with the relationship between DPCA and IDPCA technology is explained.

#### 2.1.1. The Essence of Space-Time Coupling Characteristics

It is well known that the clutter with space-time coupling characteristics is measured by the airborne multichannel radar. The Doppler frequency of clutter distribution is determined by the azimuth angle, which is called space-time coupling. The essence of space-time coupling characteristics can be obtained by analyzing the airborne multichannel radar signal model.

Before the derivation, the limited physical scene is that the multichannel airborne radar works in side-looking view, that is, the movement direction of the airborne platform is parallel to the axis of the array. As shown in Figure 1, the radar antenna is composed of $N$ elements, the distance between the arrays is $\Delta d$ (no more than half a wavelength), the radar array is shared by transmit and receive array, and the transmit and receive antenna array patterns are, respectively, $T(\theta)$ and $P(\theta)$, which can be seen by Equation (1). In Equation (1), $\lambda$ is the wavelength, $\Delta d$ is the distance between arrays, $\theta$ is the angle between the radar beam pointing and the normal direction of the antenna.

$$T(\theta) = P(\theta) = \sum_{n=1}^{N} \exp\left(j\frac{2\pi n}{\lambda}\Delta d \sin\theta\right) \tag{1}$$

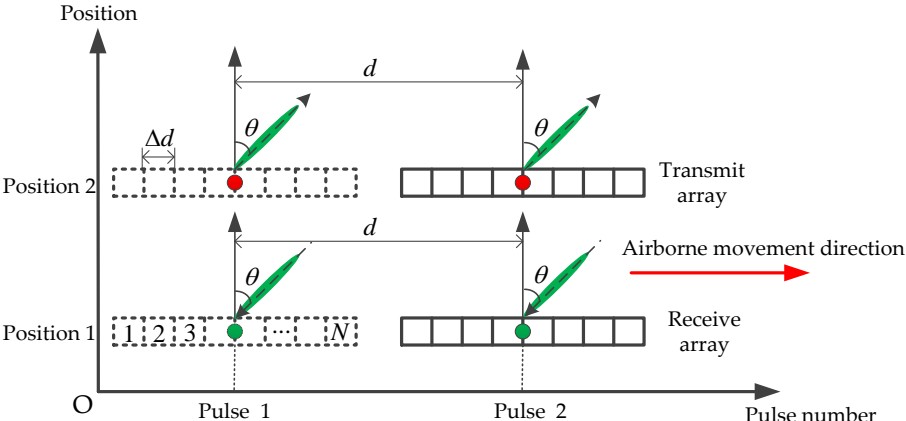

**Figure 1.** Schematic diagram of the airborne multichannel radar transmission and reception.

In the very short transceiver time, the position 1 and 2 corresponding to the transmit array and the receive array are coincident, the distance that the airborne radar moves in a pulse repetition period interval (between pulse 1 and pulse 2) is $d$, and the echo signal $s(\theta, m)$ corresponding to the $m$-th pulse is proportional to the phase difference caused by the airborne motion (see Equation (2)). The phase term $\exp(j2\pi md \sin\theta/\lambda)$ is due to the difference in path between adjacent pulses which caused by the airborne platform motion.

$$s(\theta, m) \propto \left[ T(\theta) \exp\left( j\frac{2\pi m}{\lambda} d \sin\theta \right) \right] \left[ P(\theta) \exp\left( j\frac{2\pi m}{\lambda} d \sin\theta \right) \right] \qquad (2)$$

After the Fourier transform of the $M$ pulses received in Equation (2), the spectral component $S(\theta, k)$ is shown in Equation (3).

$$S(\theta, k) = \sum_{m=1}^{M} T(\theta)P(\theta) \exp\left[ j2\pi n \left( \frac{2d}{\lambda} \sin\theta - \frac{k}{M} \right) \right] \qquad (3)$$

Using the pulse repetition frequency to normalize the Doppler frequency, the normalized Doppler spectral frequency corresponding to the clutter is shown in Equation (4), where PRF is short for Pulse Repetition Frequency.

$$f(\theta) = \frac{2d}{\lambda PRF} \sin\theta \qquad (4)$$

It can be seen from Equation (4) that the Doppler frequency of the clutter is related to the spatial azimuth angle, and the distance $d$ that the airborne platform moves in one pulse period cannot be exceed $\lambda/2$, otherwise the Doppler spectrum will appear aliasing. At the same time, compared with the phase corresponding to the first pulse, the phase of both the transmitting and receiving arrays have the same change for the airborne multichannel radar. The change in phase centers of the transmit array and receive array makes the space and time domains dependent upon each other. It can be seen, in essence, that the reason for the space-time coupling characteristics of airborne multichannel radar clutter is that the phase centers corresponding to the transmit array and the receive array in the pulse repetition interval have changed.

### 2.1.2. Relationship between DPCA and IDPCA Technology

According to the knowledge of the essence of space-time coupling characteristics, this section firstly describes the Displaced Phase Center Antenna (DPCA) technology, which can be used for compensating phase difference caused by the motion of the airborne platform. Then, with the help of the idea that the adjacent pulse phase difference can be compensated by the DPCA technology, the Inverse Displaced Phase Center Antenna (IDPCA) technology

is proposed for generating the adjacent pulse phase difference which is generated by the emulated airborne platform moving.

The core purpose of the DPCA technology is to make two adjacent pulses send out at the same position. The basic principle is that the antenna phase center moving distance can be adjusted as according to the moving distance of the airborne platform, which can be achieved by adjusting the speed of the airborne platform and the Pulse Repetition Frequency (PRF). As shown in Figure 2, if an antenna is divided into two sections of equal length, the continuous pulses are sent out alternately by the two antenna sections: the pulse $n$ is sent from the front section of the antenna, the pulse $n + 1$ is sent from the back section, the pulse $n + 2$ is sent from the front section of the antenna, and the pulse $n + 3$ is sent from the back section, and this is repeated in turn. In a pulse repetition interval, the airborne platform moves from the phase center of the back antenna to the phase center of the front antenna, and it can be known that each pair of pulses (pulse $n$ and pulse $n + 1$) will be emitted from the same position.

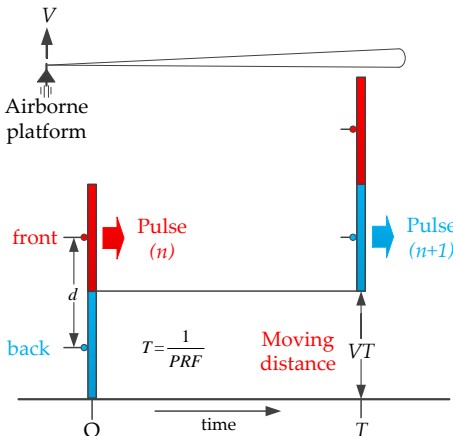

**Figure 2.** Schematic diagram of DPCA technology.

The midpoint of the phase centers of the transmitting and receiving antennas corresponding to each pair of pulses is taken as the equivalent phase center. As shown in Figure 2, the airborne multichannel radar is in the side-looking view mode, and the moving speed of the airborne platform is $V$. The phase center of the antenna corresponding to the front section is 0, after the time $T$, and the phase center of the antenna corresponding to the back section serving as the receiving antenna is $VT + d/2$. According to the definition of the equivalent phase center, the total displacement of the equivalent phase center corresponding to the pulse $n$ can be obtained as $VT + d/2$, and similarly, the total displacement of the equivalent phase center corresponding to the pulse $m$ can be obtained as $VT + d/2$. The specific analysis results are shown in Table 1.

**Table 1.** DPCA adjacent pulse phase center shift.

| Pulse | The Displacement of the Phase Center | | |
|:---:|:---:|:---:|:---:|
| | **Transmitting** | **Receiving** | **Total** |
| $n$ | $0$ | $VT + d/2$ | $VT + d/2$ |
| $n + 1$ | $d/2$ | $VT$ | $VT + d/2$ |

From the analysis results in Table 1, it can be seen that although the two adjacent pulses are not sent from the same spatial position, and the echoes of the two adjacent pulses are not received from the same spatial position, the corresponding equivalent phase center of the two adjacent pulses move by the same distance. The moving distance of the equivalent phase center corresponding to adjacent pulses is 0. Physically, it can be considered that two adjacent pulses are sent from the same point, and the echoes are received from the same

point. It shows that the DPCA technology can compensate the phase difference caused by the airborne platform motion. It should be noted that the implementation of the DPCA technology must meet the following conditions: (1) the PRF is related to the speed of the aircraft; (2) the direction of movement of the aircraft and the axis of the antenna are parallel; and (3) the phase and amplitude characteristics of the two antenna segments must be an exact match.

### 2.1.3. Implementation of IDPCA Technology for Shore-Based Multichannel Radar

The core of the DPCA technology is to compensate the phase difference caused by the motion of the airborne platform, while the IDPCA technology aims to generate the phase difference caused by the motion of the airborne platform. In this paper, the IDPCA technology is implemented by the shore-based multichannel radar with co-located transceiver antennas, i.e., an array of antennas used for both transmitting and receiving. Based on this co-located multichannel radar, the basic principle of IDPCA implementation is that the transmitting sub-array moves between pulses while the receiving array remains unchanged. It should be noted that the transmit subarray is part of the full array and the receive array is the full array. The technical details are as follows.

For the convenience of description, an array composed of 12 subarrays is taken as an example, as shown in Figure 3. Assuming that each subarray is used as a transmit subarray, the phase centers of the transmit antenna are $\varphi_{t1}, \varphi_{t2}, \cdots, \varphi_{t12}$, and the distance between the antenna phase centers is $\lambda/2$, and all the channels receive returns at the same time. Because the variance of the receive phase center of each channel between adjacent pulses is consistent with the variance of the receive phase center of the whole array, for simplification, the whole array is used to receive the signal, and the phase center of the receive antenna is $\varphi_R$. The equivalent phase centers of transmit and receive are approximately equal to the midpoint position between the transmit centers and receive phase centers. According to this principle, the equivalent phase center of the first transmit subarray corresponding to the first pulse is $\varphi_{Rt1}$, and the equivalent phase center of the second transmit sub-array corresponding to each pulse is $\varphi_{Rt2}$; see Equation (5).

$$\varphi_{Rt1} = \frac{\varphi_R - \varphi_{t1}}{2} = \frac{5.5\lambda}{4}, \; \varphi_{Rt2} = \frac{\varphi_R - \varphi_{t2}}{2} = \frac{4.5\lambda}{4} \tag{5}$$

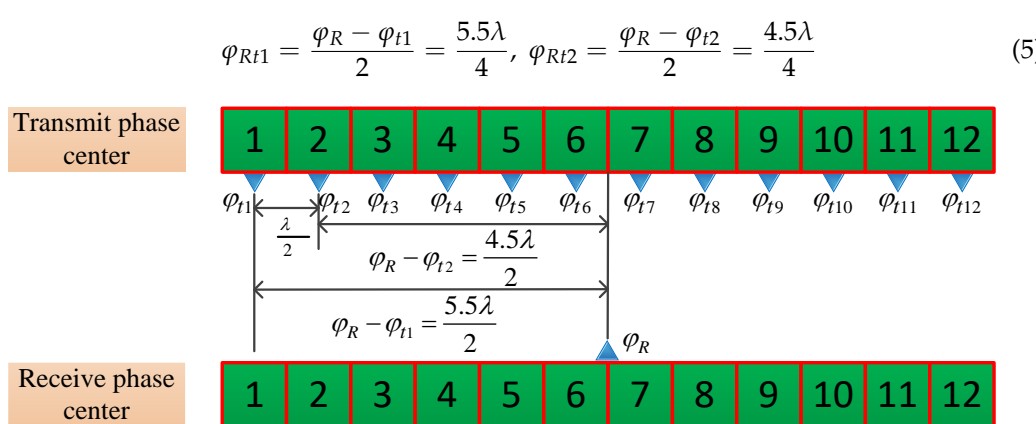

**Figure 3.** Schematic of transmit and receive phase centers.

Then, the transmit subarray moves one channel interval $\lambda/2$ with the pulse repetition interval $PRI$, so the moving distance of the equivalent phase center of two adjacent transmit subarray is $\Delta\varphi$, as shown in Equation (6).

$$\Delta\varphi = \varphi_{Rt1} - \varphi_{Rt2} = \frac{\lambda}{4} \tag{6}$$

By setting the value of $PRI$ and the moving distance of the equivalent phase center, the speed of emulating aircraft motion can be obtained, that is $v = \Delta\varphi/PRI$.

According to the implementation of IDPCA technology for shore-based multichannel radar, it can be seen that there are three features in the aircraft motion emulation. First,

since the moving direction of the aircraft motion is always perpendicular to the direction of the transmit antenna, the emulate aircraft motion is equivalent to the airborne multi-channel radar working in side-looking mode. Second, because the distance and time of the movement between pulses are the same, the speed of aircraft motion emulation is constant. Third, the speed of aircraft motion emulation is related to the moving distance of the adjacent transmit sub-arrays and PRI.

### 2.2. The Measurement System of Multichannel Sea Clutter

Section 2.1 gives a detail introduction to the essence of space-time coupling characteristics and the techniques for generating space-time coupling characteristic data. Based on the above theoretical knowledge, this section introduces the sea clutter measurement system of the emulated airborne movement, which is implemented based on the shore-based multichannel radar.

### 2.2.1. The Measurement System

It is well known that the essence of the space-time coupling is the phase difference caused by the airborne movement. Figure 4a shows a Cartesian coordinate system as $O - xyz$, where $y$-axis denotes the platform moving direction, and $z$-axis denotes perpendicular to the sea surface. In addition, $H$ denotes the height of radar, $\theta$ and $\varphi$ denote the elevation angle and azimuth angle of the sea surface azimuth patch relative to the transmit sub-array, respectively, M denotes the number of pulses included in single CPI, and N denotes the number of the array channels.

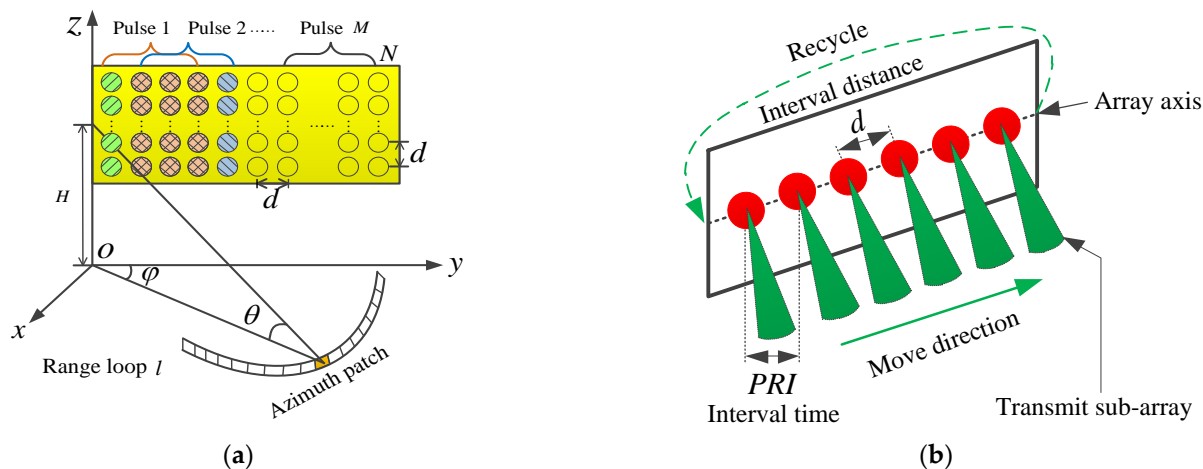

**Figure 4.** The sketch map of multichannel sea clutter measurement system: (**a**) transmit and receive array; (**b**) transmit subarray moves cyclically as the CPI changes.

In order to generate the phase difference, the measurement system needs to meet the following conditions. First, the radar must be multichannel system to obtain the space information. As shown in Figure 4a, the radar using a planar array antenna with a total number of 12 channels, which is located on the island. Second, the transmit subarray moves at equal intervals between pulses, and all the channels receive signals at the same time. In this paper, the transmit subarray is composed of four channels, and each subarray correspond to one channel. In addition, the distance moved between the adjacent subarrays is $d$ (no more than half a wavelength). For example, the transmit subarrays corresponding to the first pulse are numbered 1–4, and the transmit subarrays corresponding to the second pulse are numbered 2–5, and the last transmit subarrays corresponding to the ninth pulse are numbered 9–12. Third, the transmit subarray moves periodically with the pulse repetition period, as shown in Figure 4b. For example, when the transmit subarray moves from the head of the array axis to the tail of the array axis, a coherent processing interval (CPI) is completed in the pulse dimension, then the transmit subarray moves again from

the head to the tail of the array, and the transmit subarray moves periodically until shaping multiple CPIs.

According to the above three conditions, which are satisfied by the multichannel radar used for emulating the airborne platform movement, the sea clutter measurement system has the following properties. (1) The transmit subarray used for moving and the receive array share the same array, and the transmit subarray is a part of the entire array. So, the pulse number of the CPI is determined by the total number of channels and the number of channels which contained by the transmit subarray. (2) The data structure of a single channel is illustrated by Figure 5: *M* columns corresponding to *M* pulse repetition interval, *L* lines corresponding to *L* range samples, and there are multiple CPIs in pulse dimension. Therefore, the two-dimensional matrix is composed of *L* range gates and *MK* pulses, which is shown as Figure 5. If combing *N* channels, the two- dimensional matrix is raised to three-dimensional. (3) For clutter data matrix consisting of single CPI and multiple range gates, the data information of *L* range gates in one pulse are the various motion postures of the sea surface at this instant. The velocity of the sea surface movement for each range gate can be obtained from the multiple pulses in the single CPI. (4) For clutter data matrix consisting of multiple CPIs and single range gate, there is no spatial and temporal correlation between CPIs. It means that multiple CPIs data are multiple independent repeated measures experiments for a single range gate. Multiple CPIs data are equivalent to taking multiple photos of the sea surface in the single range ring, including various motion postures of the sea surface in multiple moments.

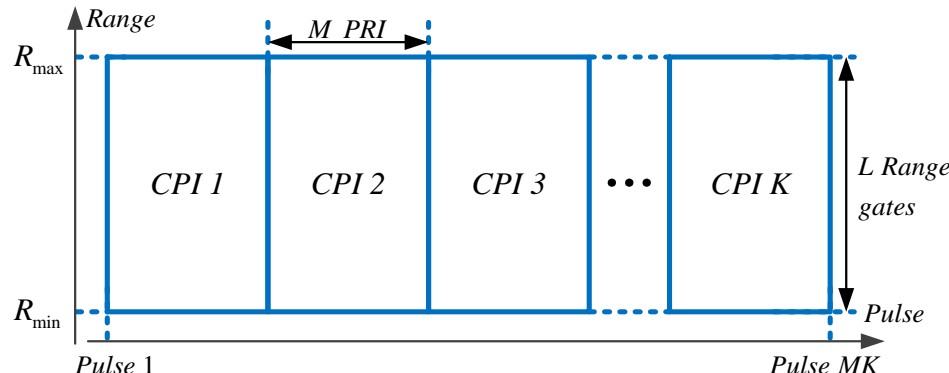

**Figure 5.** The single channel data matrix (slow time × fast time).

2.2.2. Space-Time Signal Model of Sea Clutter in a Single Range Cell

The geometry of the clutter patch is depicted in Figure 4a, which the range ring is composed of many azimuth patches. In the side-looking configuration, the movement of the emulated platform is aligned along with the positions of the receive channels. The pulse repetition frequency (PRF) is denoted as $f_r = 1/T_r$, where $T_r$ is the pulse repetition interval (PRI). Within each PRI, there are $L$ time (range) samples. $v_{ci}$ is the radial velocity of the sea clutter patch at the $i$-th pulse corresponding to the slow time $i \cdot T_r$, which varies with pulses in the same position. The distance moved between the subarrays is $d$, and $\bar{v}_c$ is the average velocity of the sea clutter patch for $M$ pulses. The slant range at the $m$-th pulse with a specific direction of arrival $(\varphi, \theta)$ is expressed as in Equation (7).

$$
\begin{aligned}
R((m-1)T_r) &= vmT_r \cos\varphi\cos\theta + (v_{c1}T_r + v_{c2}T_r + \cdots + v_{cm}T_r) + dm\cos\varphi\cos\theta \\
&= vmT_r\cos\varphi\cos\theta + \begin{bmatrix} (v_{c1} - \bar{v}_c + \bar{v}_c)T_r + (v_{c2} - \bar{v}_c + \bar{v}_c)T_r \\ + \cdots + (v_{cm} - \bar{v}_c + \bar{v}_c)T_r \end{bmatrix} + dm\cos\varphi\cos\theta \\
&= (v\cos\varphi\cos\theta + \bar{v}_c)mT_r + \left( \sum_{i=1}^{m} v_{ci} - m\bar{v}_c \right)T_r + dm\cos\varphi\cos\theta \\
&= R_t + R_s
\end{aligned}
\tag{7}
$$

The aforementioned slant range consists of the temporal part $R_t$ and spatial part $R_s$. The temporal angle frequency is determined by the temporal part and is written as in Equation (8).

$$
\begin{aligned}
\omega_m &= 2\pi \frac{2v_{all}}{\lambda f_r} = \frac{4\pi}{\lambda} R_t \\
&= 2\pi \left( \frac{2v}{\lambda f_r} \cos\varphi \cos\theta + \frac{2\overline{v}_c}{\lambda f_r} \right)(m-1) + 2\pi \frac{2\left( \sum\limits_{i=1}^{m} v_{ci} - k\overline{v}_c \right)}{\lambda f_r}
\end{aligned}
\tag{8}
$$

where $v_{all}$ is the velocity which caused the Doppler frequency shift. The temporal steering vector of a fluctuating point source for the $l$-th range cell is given as

$$
\mathbf{S}_t(\theta, \varphi) = \left[ 1, e^{j\omega_1}, \cdots, e^{j\omega_M} \right]^T
\tag{9}
$$

where superscript $[]^T$ denotes the transpose. The spatial steering vector is written as

$$
\mathbf{S}_s(\theta, \varphi) = \left[ 1, e^{j2\pi \frac{d}{\lambda} \cos\theta \cos\varphi}, \cdots, e^{j2\pi \frac{(N-1)d}{\lambda} \cos\theta \cos\varphi} \right]^T
\tag{10}
$$

where $(d/\lambda)\cos\theta\cos\varphi$ denotes the spatial frequency. Therefore, the space-time steering vector is written as

$$
\mathbf{v}(\theta, \varphi) = \mathbf{s}_t(\theta, \varphi) \otimes \mathbf{s}_s(\theta, \varphi)
\tag{11}
$$

where $\otimes$ is the Kronecker.

The radar space-time echo of the $l$-th range cell at the $m$-th pulse is the sum of the $P$ clutter patches from different azimuth angles and is given as

$$
\mathbf{x}_{l,m} = \sum_{p=1}^{P} \xi_m(\theta_l, \varphi_p) \cdot e^{j\omega_m} \cdot \mathbf{s}_s(\theta_l, \varphi_p)
\tag{12}
$$

where $\xi_m(\theta_l, \varphi_p)$ is the amplitude of the clutter patch at the $m$-th pulse computed from the radar equation.

$$
\xi_m(\theta_l, \varphi_p) = \left( \frac{P_t G_t(\theta_l, \varphi_p) G_r(\theta_l, \varphi_p) \lambda^2 \sigma_m(\theta_l, \varphi_p)}{(4\pi)^3 N_o L_s R_m^4} \right)^{1/2}
\tag{13}
$$

where $P_t$ is the transmit power, $G_t(\theta_l, \varphi_p)$ is the sub-array transmit power gain of the elevation azimuth point $(\theta_l, \varphi_p)$, $G_r(\theta_l, \varphi_p)$ is the receive element gain of the elevation azimuth point $(\theta_l, \varphi_p)$, $N_o$ is the noise power, $L_s$ is the system losses, $R_m$ is the slant range of the $m$-th pulse, and $\sigma_m(\theta_l, \varphi_p)$ is the $m$-th pulse effective radar cross section (RCS) of the elevation azimuth point $(\theta_l, \varphi_p)$, which is given in Equation (14).

$$
\sigma_m(\theta_l, \varphi_p) = \sigma_o(\theta_l, \varphi_p) R_m \Delta\varphi_p \Delta R \sec^2\theta_l
\tag{14}
$$

where $\sigma_o(\theta_l, \varphi_p)$ is the backscattering coefficient of sea clutter, $\Delta\varphi_p = 2\pi/P$ is the minimum azimuth unit of division, where $P$ denotes the number of the clutter azimuth patches, and $\Delta R$ is the radar radial range resolution. It can be easily concluded that the scattering coefficient of the sea clutter corresponding to each azimuth patch is different. Therefore, the amplitude of the specific clutter patch varies with the pulse. Then the space-time sea clutter is given as

$$
\mathbf{x}_l = \left[ \mathbf{x}_{l,1}^T, \cdots, \mathbf{x}_{l,m}^T, \cdots, \mathbf{x}_{l,M}^T \right]^T
\tag{15}
$$

Compared with the space-time signal model of the land clutter in a single range cell, there are some obvious differences existed in magnitude and phase of the space-time signal model of the sea clutter in a single range cell. The differences existed in the magnitude are caused by the variant scattering coefficient of sea clutter [21]. Moreover, the variation properties of the sea clutter are reflected in two levels, the first level is the scattering

coefficient of sea clutter, varying with the different azimuth patches which belong to the single range cell, and the other level is the scattering coefficient of sea clutter, varying with the different pulses which belong to the single range cell. The main reason for the difference of the former level is the difference caused by the angle between the wave direction and the radar line of sight, and the main reason for the difference of the latter level is the pulse-to-pulse change of the wave direction which caused by the motion of the sea surface. The differences exist in the phase caused by the sea clutter intrinsic motion, which is obviously reflected in the temporal steering vector. It can be seen that the temporal steering vector is composed of two aspects, which can be seen in Equation (16). One is the traditional temporal steering vector $\mathbf{s}_{to}$ caused by the movement of the radar platform, the other is the temporal steering vector $\mathbf{s}_{tc}$ caused by the intrinsic motion of the sea clutter. It should be noted that the average velocity of the sea clutter for $M$ pulses $\bar{v}_c$ and sea surface radial velocity $v_{ci}$ vary with sea states, where $\odot$ denotes the Hadamard product.

$$
\begin{aligned}
\mathbf{s}_t &= \left( e^{j2\pi\left(\frac{2v}{\lambda fr}\right)\cos\theta\cos\varphi(m-1)} \right)_{m=1}^{M} \odot \left( e^{j\frac{2\pi}{\lambda fr}\left(2(m-1)\bar{v}_c + 2\left(\sum\limits_{i=1}^{m} v_{ci} - k\bar{v}_c\right)\right)} \right)_{m=1}^{M} \\
&= \mathbf{s}_{to} \odot \mathbf{s}_{tc}
\end{aligned}
\tag{16}
$$

### 2.3. The Method of the Measured Data Processing

Compared with the ground clutter, the particularity of sea clutter space-time characteristics is the variability of the scattering coefficient of the sea clutter and the speed of the sea surface. Furthermore, the scattering coefficient of the sea clutter and the speed of the sea surface vary with the parameters which used for describing sea states. For the clutter data in Section 2.2.2, the scattering coefficient of the sea clutter and the speed of the sea surface vary with time in pulse dimension, and the scattering coefficient of the sea clutter varies with the grazing angles in range dimension. However, the sea clutter is modulated by the radar antenna pattern, range attenuation factor, and different clutter patch areas. The sea clutter data shows inhomogeneity in range dimension. In order to analyze the effects of the scattering coefficient of the sea clutter and the speed of sea surface on the space-time characteristics of sea clutter, this section reduce or compensate for these effects next.

In accordance with the Figure 6, the method of the measured data processing is divided into three steps: sea clutter data collection, the intensity of sea clutter compensation and the space-time clutter processing.

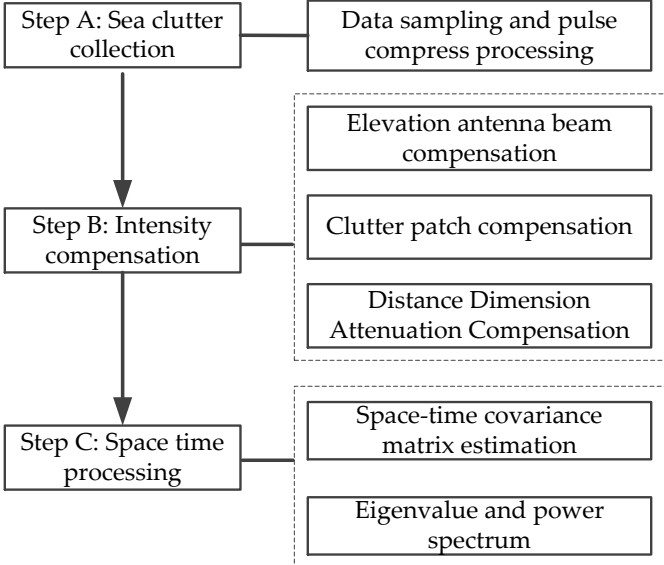

**Figure 6.** Flow chart of the method of multichannel radar sea clutter data processing.

Step A: sea clutter data collection.

According to the Ward [25], each receive channel has its own down converter, matched filter receiver, and A/D converter. For each PRI, $L$ range samples are collected to cover the range interval. With $M$ pulses and $N$ receiver channels, the received data for one CPI comprises $LMN$ complex samples. This assembly will be referred to as the CPI datacube which is a three-dimensional data matrix. Most of the radar signals are linear frequency modulated signals. In order to improve the range resolution, the range dimension data need to be processed by pulse compression processing.

Step B: Intensity compensation.

Based on radar Equation (12), there are three factors that affect the amplitude of sea clutter per range cell. First, the gain for the antenna pattern of the elevation beam varies with the grazing angles. Second, according to Equation (14), the effective RCS (Radar Cross Section) of the clutter patch is determined by the backscattering coefficient of sea clutter $\sigma_o(\theta_l, \varphi_p)$ and the area of the clutter patch which varies with the range cell. It is noted that the backscattering coefficient of sea clutter is regarded as the result of the resonance between the incident wave and the sea surface wave, and the backscattering coefficient of sea clutter which belongs to the different area cannot be compensated. Third, the amplitude of sea clutter is attenuated with the number of range cell increases. We can compensate for the inhomogeneity of the amplitude caused by the above three factors.

(1) The gain compensation of the elevation antenna pattern.

Taking the maximum value of the elevation antenna gain as a reference, the elevation antenna gain value corresponding to each range gate can be compensated, which is the basic idea for the gain compensation of the elevation antenna pattern.

(a) According to the radar height $h_{radar}$ and the actual distance $r$ corresponding to the range cell $i$, calculating the elevation angle $\theta_i$ of the radar beam center corresponding to the $i$-th range cell, see Equation (17).

$$\theta_i = \arcsin\left(\frac{h_{radar}}{r}\right) \tag{17}$$

(b) Based on the gain function of the elevation antenna pattern $g(\theta)$, finding the difference $\Delta g_i$ between the pattern gain corresponding to the elevation angle $\theta_i$ and the maximum gain $g_{\max}(\theta)$ of the elevation antenna pattern can be seen in Equation (18). For the echo power of each elevation angle in each channel, multiply the echo power of the range cell by the two-way gain compensation value $-2\Delta g_i$.

$$\Delta g_i = g(\theta_i) - g_{\max}(\theta) \tag{18}$$

(2) The area compensation of the clutter patch.

According to Equation (14), the differences between the area of the clutter patches are the radial range and the grazing angle corresponding to each range cell. Taking the radial range $r_o$ and the grazing angle $\theta_o$ belonging to the center of the elevation angle as the reference, the area of different clutter patch can be compensated. Firstly, by calculating the area of the reference clutter patch $S_o$ based on Equation (19). Secondly, by calculating the ratio $\Delta S_i$ of the clutter patch area belonging to each range cell to the reference patch area. At last, by compensating the area of the clutter patch in each range cell with the area ratio $\Delta S_i$.

$$S_o = r_o \sec^2 \theta_o$$
$$\Delta S_i = S_i / S_o \tag{19}$$

(3) The range dimension attenuation compensation.

The method of range dimension compensation is similar to the above two methods. The range gate corresponding to the center of the elevation beam is selected as the reference gate, and the other range gates perform range-dimension attenuation compensation according to the radar equation. Firstly, assuming that the distance corresponding to the center of the elevation beam is $R_c$, calculate the ratio of other distances to the corresponding ratio $\Delta_i$,

which is seen in Equation (20). Secondly, for the echo power $P_i$ corresponding to the range $R_i$, the compensated power $P_i'$ is shown in Equation (20). Thirdly, according to the above algorithm, the range dimension power compensation is performed on each channel.

$$\Delta_i = R_i / R_c$$
$$P_i' = P_i \Delta_i{}^4 \tag{20}$$

In order to more intuitively illustrate the effect before and after sea clutter amplitude compensation, Figure 7 takes the one-dimensional echo sequence corresponding to a single channel as an example, and the amplitudes of the one-dimensional echo sequence are sequentially performed the above compensation. In Figure 7, compared with the one-dimensional amplitude of single channel each range gate, the amplitude difference between different range gates become smaller after the amplitude compensation of the elevation antenna pattern and area of the clutter patch, and then become even smaller when the range dimension attenuation is compensated.

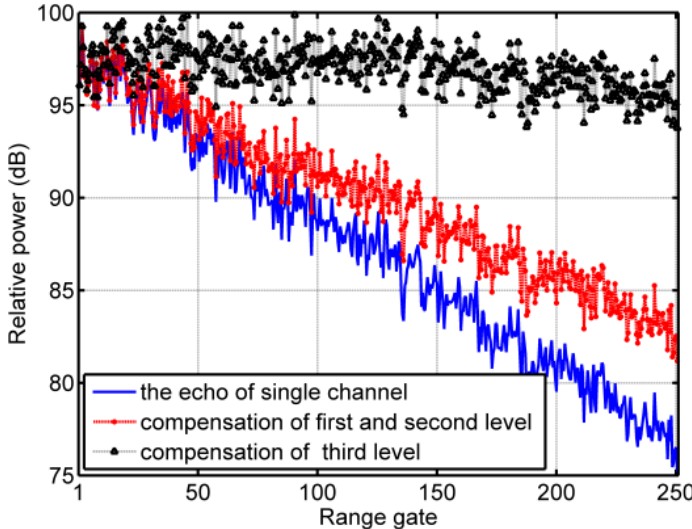

**Figure 7.** Example of the echo amplitude before and after echo amplitude is compensated.

Step C: Sea clutter space-time processing.

In accordance with Section 2.2.2, the structure of the measured single channel matrix is shown as Figure 5. There are multiple CPIs in pulse dimension, and there is no correlation between individual CPIs. For the measured data of each single channel, the data in pulse dimension can be divided multiple groups of coherent pulse trains. The smallest unit of the space-time processing is the space-time snapshot of a single range cell. In order to obtain the space-time snapshot, it is necessary to slice along the range dimension perpendicular to the three-dimensional data matrix.

It is known that the estimation of the space-time covariance matrix is crucial for the analyzation of the sea clutter space-time characteristics. The number of samples to estimate the covariance matrix needs to satisfy the RMB criterion [25], which is that the number of the sample is twice the space-time degrees of freedom. The RMB criterion is carried out under the premise that the samples satisfy the independent and identical distribution in the statistical sense. The space-time covariance matrix is estimated using the matrix composed of a single space-time snapshot and multiple range cells traditionally. However, the space-time snapshots corresponding to each range cell are not independent and identically distributed, especially for the sea clutter. According to Equation (14), the backscattering coefficient of sea clutter corresponding to each range cell is different, and this effect is not compensated.

In this paper, the space-time covariance matrix of sea clutter is estimated using the matrix composed of a single range cell and multiple space-time snapshots. This covariance

matrix estimation method avoids the problem of the scattering coefficient of sea clutter varies with the range cells. According to the maximum likelihood method, the space-time covariance matrix can be estimated with Equation (21), where $H$ is the conjugate transpose operation, $Q$ is the number of the CPI groups, and $\mathbf{x}_q$ is the space-time snapshot belongs to the $q$-th CPI.

$$\hat{\boldsymbol{R}}_c = \frac{1}{Q}\sum_{q=1}^{Q} \mathbf{x}_q \mathbf{x}_q^H \tag{21}$$

It can be seen from Equation (19) that the space-time covariance matrix $\hat{\boldsymbol{R}}_c$ is a Hermite matrix, which can be spectrally decomposed. In Equation (22), $\lambda(m)$ is the $m$-th eigenvalue belongs to the $m$-th eigenvector, and all the eigenvalues are greater than 0. If the eigenvalues are arranged in order, the eigenvalue spectrum of the space-time covariance matrix is formed.

$$\hat{\boldsymbol{R}}_c = \sum_{m=1}^{NM} \lambda(m)\boldsymbol{p}(m)\boldsymbol{p}^H(m);$$
$$\lambda(1) \geq \lambda(2) \geq \cdots \geq \lambda(NM) > 0 \tag{22}$$

Using Equation (23) to estimate the Capon space-time power spectrum [25], which can intuitively reflect the optimal performance of clutter suppression,

$$P(\vartheta,\varpi) = \frac{1}{\mathbf{v}(\vartheta,\varpi)^H \hat{\boldsymbol{R}}_c^{-1} \mathbf{v}(\vartheta,\varpi)} \tag{23}$$

where $\mathbf{v}(\vartheta,\varpi)$ is the $NM \times 1$ dimensional space-time steering vector, shown as Equation (11), and $\hat{\boldsymbol{R}}_c^{-1}$ is the sample matrix $\hat{\boldsymbol{R}}_c$ inversion.

## 3. Results

In order to verify the validity of the measurement data in the multichannel sea clutter measurement system in Section 2.2.1, firstly, this section simulates multichannel sea clutter based on the multichannel radar sea clutter model in Section 2.2.2, secondly, the eigenvalue spectrum and space-time power spectrum are estimated with the method in Section 2.3, then the validity of the data measured by the multichannel sea clutter measurement system is compared and analyzed. In the end, the results of measured sea clutter space-time processing are given.

### 3.1. Validation Results from the Measurement and Space-Time Processing Method

3.1.1. Method of the Validation

Guided by Section 2.2.2, this paper applies the existing sea clutter amplitude distribution model and power spectrum model to the multichannel sea clutter simulation. The simulation is carried out at three levels.

First, simulate the intensity corresponding to each sub-patch in the clutter range ring. According to Equation (12), the multichannel sea clutter corresponding to a single range cell is formed by the sum of the echo amplitudes of each sub-patch. Therefore, it is necessary to determine which distribution the amplitude corresponding to each range cell obeys, and which distribution the amplitude corresponding to each azimuth sub-patch obeys. For the distribution of the range cell amplitude, it has been found that sea clutter can be modeled by a compound random process in which the speckle whose amplitude $x$ is modulated by an underlying mean intensity $y$ (texture) [21]. The speckle comes from locally Gaussian random process, and a correlated gamma distribution is often used to model the texture, yielding a clutter model whose amplitude is K-distributed, which can be seen in Equation (24). The shape parameter $v$ and the scale parameter $b$ can be estimated by an empirical model and the mean power of the clutter in a resolution cell [25]. For the distribution of each azimuth sub-patch, there is no explicit model. For

reference [20], the two-scale backscattering coefficient model can be used to calculate the specific backscattering coefficient of sea clutter at different time and positions.

$$p(x|y) = \frac{2x}{y} \exp\left(-\frac{x^2}{y}\right)$$
$$p(y) = \frac{b^v}{\Gamma(v)} y^{v-1} \exp(-by) \tag{24}$$
$$p(x) = \int_0^\infty p(y)p(x|y)dy = \frac{4b^{(v+1)/2}}{\Gamma(v)} x^v \mathrm{K}_{v-1}\left(2x\sqrt{b}\right)$$

Second, simulate the coherent time series. The Doppler spreading of the speckle comes from the internal motion of the sea clutter. A Gaussian shape can model the power spectral density (PSD) in Equation (25), where $f$ is the Doppler frequency, $y$ is the texture, and $\delta$ is the standard deviation of the PSD. For each clutter sub-patch, generate a complex spectrum with the appropriate mean intensity but with the same normalized power spectral density, with random values of speckle in each sub-patch. It has been found that the mean Doppler frequency $m_f(y)$ varies with the underlying intensity. The spectrum width $\delta$ is Gaussian distribution. The estimation of the above two parameters can be found in the reference [26,27].

$$G(f,y,\delta) = \frac{y}{\sqrt{2\pi}\delta} \exp\left(-\frac{\left(f - m_f(y)\right)^2}{2\delta^2}\right) \tag{25}$$

Third, simulate coherent clutter on multichannel. For each sub-patch, the contribution to its phase comes from two parts: one is the velocity phase with respect to the Doppler shifted according to the sub-patch's angular position, the other is the spatial phase with respect to the position of each phase center of the related sub-array. The velocity phase can be seen the exponential terms $y_{1,l,m}$ in Equation (26), where 1 denotes 1-th channel, $l$ denotes $l$-th range cell, and $m$ denotes $m$-th pulse. The spatial phase can been the second exponential terms $y_{2,l,m}$ in Equation (26), which is related to the distance between reference sub-array and current sub-array, where $d$ is the distance between two adjacent sub-array, $PRI$ is the pulse repetition interval.

$$
\begin{aligned}
y_{1,l,m} &= \sum_{k=1}^{P} y_m(\theta_l, \varphi_k) \exp\left(-j\frac{4\pi vmPRI}{\lambda} \cos\theta_l \cos\varphi_k\right) \\
y_{2,l,m} &= \sum_{k=1}^{P} y_m(\theta_l, \varphi_k) \exp\left(-j\frac{4\pi vmPRI}{\lambda} \cos\theta_l \cos\varphi_k\right) \exp\left(\frac{2\pi d \cos\theta_l \cos\varphi_k}{\lambda}\right) \\
&\vdots \\
y_{N,l,m} &= \sum_{k=1}^{P} y_m(\theta_l, \varphi_k) \exp\left(-j\frac{4\pi vmPRI}{\lambda} \cos\theta_l \cos\varphi_k\right) \exp\left(\frac{2\pi(N-1)d \cos\theta_l \cos\varphi_k}{\lambda}\right)
\end{aligned}
\tag{26}
$$

In summary, the flow of the strategy which used for simulating one CPI data matrix is clearly summarized as the algorithm, which is shown in Table 2. The data matrix of sea clutter whose form, as shown in Figure 5, can be generated by using the above algorithm more times.

**Table 2.** The specific steps of multichannel sea clutter simulation.

| Algorithm | Simulation of Multichannel Sea Clutter |
|---|---|
| Input | $P_t, G_t(\theta, \varphi), G_r(\theta, \varphi), \lambda, N_o, L_s$ according to Equation (11). |
| Step 0 | Divide the $l$-th range ring into $P$ sub-patches and $l = 1, \cdots, L$, where the center of each patch subtends an angle $\varphi_k$ at the center of the array and $k = 1, \cdots, P$. Select $l$-th range gate, calculate the slant range $R_l$ subtends the $l$-th range gate. |
| Step 1 | Then, calculate the effective RCS of the $k$-th sub-patch $\sigma_k(\theta_l, \varphi_k)$ according to Equation (14). |

**Table 2.** *Cont.*

| Algorithm | Simulation of Multichannel Sea Clutter |
|---|---|
| Step 2 | Calculate the mean intensity of the return in each clutter sub-patch according to Equation (13). |
| Step 3 | Generate a complex spectrum with the same normalized power spectral density according to Equation (25). Then, transform the complex spectrum into time domain, giving $M$ values of a complex time domain series $y_m(\theta_l, \varphi_k)$, where $m = 1, \cdots, M$. |
| Step 4 | For each channel, apply the appropriate phase weightings to the complex time domain series $y_m(\theta_l, \varphi_k)$, and sum over all sub-patches to $y_{n,l,m}$ in Equation (26), where $n = 1, \cdots, N$ denotes channel number. |
| Step 5 | Generate normalized K-distributed data according to Equation (24), then multiply the normalized K-distributed data with the range dimension data. |
| Step 6 | Add thermal noise independently to each channel. |
| Output | Shaping three dimension data matrix, consisting of $N$ channels, $L$ range gates and $M$ pulses. |

3.1.2. Results of the Validation

In order to validate the effectiveness of the multichannel sea clutter measurement method, this section compares the results of the measured and simulated data.

In generally, the sea clutter data is measured under sea state 2. The multichannel radar parameters are shown in Table 3, and the sea state 2 parameters of measured clutter data are shown in Table 4. The working mode of the measurement system, introduced in Section 2.2.1, is that the transmit subarray composed of four subarrays moves one channel with the PRI. The radar parameters and marine environment parameters used in the simulated multichannel parameters are consistent with parameters corresponding to the measured data.

**Table 3.** The multichannel sea clutter measurement radar parameters.

| Parameters | Value |
|---|---|
| Channel number | 12 |
| Altitude | 478 m |
| Polarization | VV |
| Pulse Repetition Frequency | 2000 Hz |
| Range bandwidth | 10 MHz |
| Pulse width | 5 μs |
| Pulse number in one CPI | 9 |
| Transmit sub-array number | 4 |
| Transmitting power | 4 kW |
| Antenna gain (transmitting) | 20.2 dBi |
| Antenna gain (receiving) | 14.2 dBi |
| Elevation angle | −4 degrees |
| Azimuth beamwidth | 23.4 degrees |
| Elevation beamwidth | 6.8 degrees |

**Table 4.** The sea state parameters of the measured data.

| Sea State | SWH (m) | Wave Direction (°) | Radar Direction (°) | Wind Speed (m/s) | Wind Direction (°) |
|---|---|---|---|---|---|
| 1 | 0.21 | 137.8 | 130 | 5.3 | 152.1 |
| 2 | 0.43 | 120.1 | 130 | 6.4 | 162.5 |
| 3 | 1.27 | 132.1 | 130 | 9.9 | 149.3 |
| 4 | 1.63 | 123.5 | 130 | 13.1 | 159.2 |
| 5 | 2.98 | 102.6 | 130 | 15.2 | 149.4 |

The eigenvalue of the measured and simulated data is shown in Figure 8, in which the eigenvalue of the simulated data comes from two kinds of processing method. The main

point of difference between the two methods are the data used in estimating the covariance matrix. The data used in the first method are composed of single CPI and multiple range gates, while the data used in the second method are composed of single range gate and multiple CPIs. It can be seen that the variation range of eigenvalues corresponding to the first method is larger than the variation range of eigenvalues corresponding to the second method. It means that estimating the covariance matrix with the multiple range gates would average the backscattering coefficient of sea clutter belonging to different grazing angle. The backscattering coefficient of sea clutter varies with the grazing angle, especially in low grazing angle. Therefore, the eigenvalue can be obtained accurately by using single range gate and multiple CPIs.

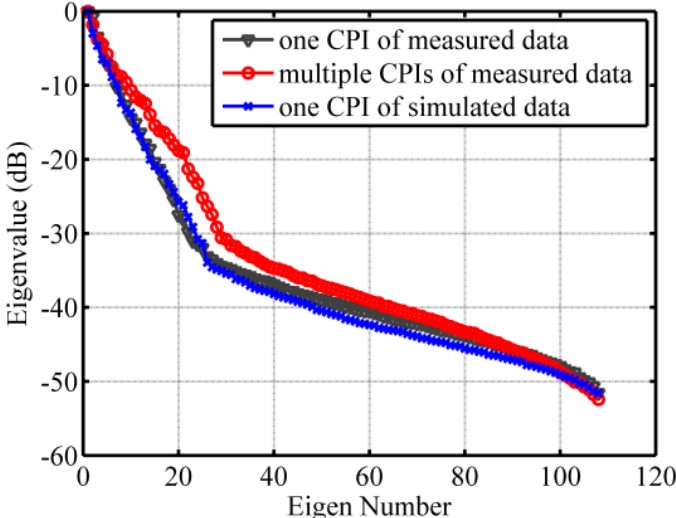

**Figure 8.** Eigenvalue spectrum.

The same conclusion can be found in the space-time power spectrum of sea clutter. The shape of sea clutter space-time power spectrum corresponding to the first method is more uniform than that corresponding to the second method, which is shown in Figure 9. There is much more information of sea clutter contained in multiple range gates than in single range gate. If the space-time covariance matrix is estimated with the multiple range gates, the inhomogeneity of sea clutter is averaged. In other words, the backscattering coefficient of sea clutter from different azimuth angles in a single range gate has little influence on the space-time power spectrum compared with the averaged backscattering coefficient of sea clutter from different range gates.

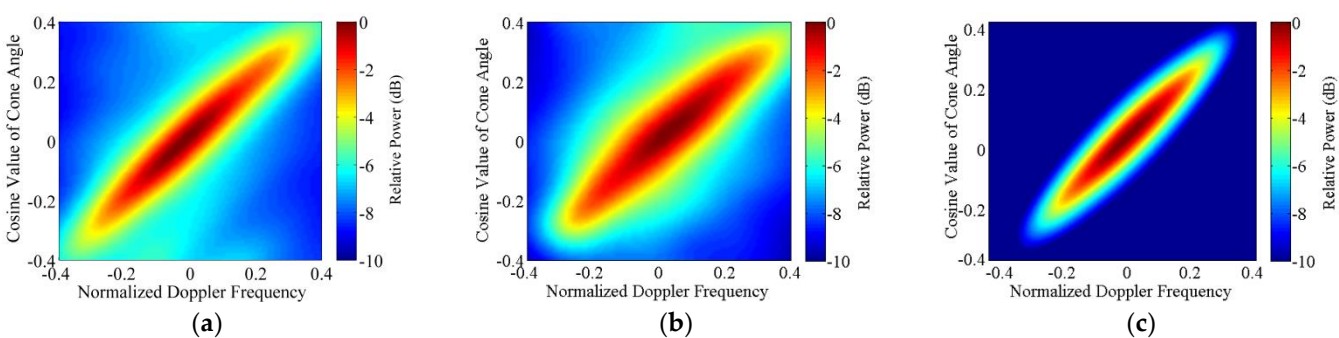

**Figure 9.** Space-time power spectrum: (**a**) measured data using single range gate and multiple CPIs; (**b**) measured data using multiple range gates and single CPI; and (**c**) simulated data using single range gate and multiple CPIs.

The comparison of Figure 9a with Figure 9c indicates that the spread of the clutter spectrum obtained by measured data is nearly the same as that obtained by the simulated data. It means that the sea clutter data collected by multichannel sea clutter measurement can better reflect the space-time characteristics of sea clutter with the processing method of single range gate and multiple CPIs.

*3.2. The Results of Measured Sea Clutter Space-Time Processing*

The multichannel sea clutter measurement experiment was conducted from 2020 to 2021 in the central Yellow Sea of China. During the experiment, a L-band multichannel radar was installed on a cliff with an altitude of 478 m. The radar parameters are shown in Table 3. It should be noted that the real measured data are all processed by amplitude and phase correction. The measurement system can emulate the airborne platform motion, is introduced in Section 2.2.1, and its working mode is that the transmit subarray composed of four subarrays moves one channel with the PRI.

It is known that the sea clutter has variant radial velocity and backscattering coefficient of sea clutter with the motion of the sea surface. Moreover, the motion of the sea surface varies with the change of the sea states. In order to compare and analyze the influence of backscattering coefficient of sea clutter and sea surface velocity on the spatial-temporal characteristics of sea clutter, we selected the sea clutter data under five sea states in this section, where the sea states are classified by the Douglas scale [28]. The related parameters of the five sea states are shown in Table 4. During the sea clutter measurement experiment, Datawell Waverider 4 provided the information of significant wave height (SWH), wave direction and cross-zero period with a data rate of a sample per 30 min, and Lufft WS700-UMB also measured real-time wind speed and wind direction with a data rate of a sample per 5 min.

For the sea clutter data under each sea state, the space-time data to be processed consisted of a single range gate and multiple pulses which contained 200 groups CPI. Based on the processing method in Section 2.3, the space-time power spectrum and eigenvalue were calculated under three range gates where the number is 1400 and 800 (corresponding to the grazing angle are 5°, 2.4° and 5°). Comparing the eigenvalue and space-time power spectrum corresponding to different range gates under each sea state in Figure 10, the range of eigenvalue and the spread width of the clutter ridge vary greatly with the different range gates. Comparing the eigenvalue and space-time power spectra under different sea states corresponding to the same range gate in Figure 10, the eigenvalue and space-time spectrum corresponding to the same range gate show the differences with the change of sea states. The reasons for the above phenomena are analyzed in the following.

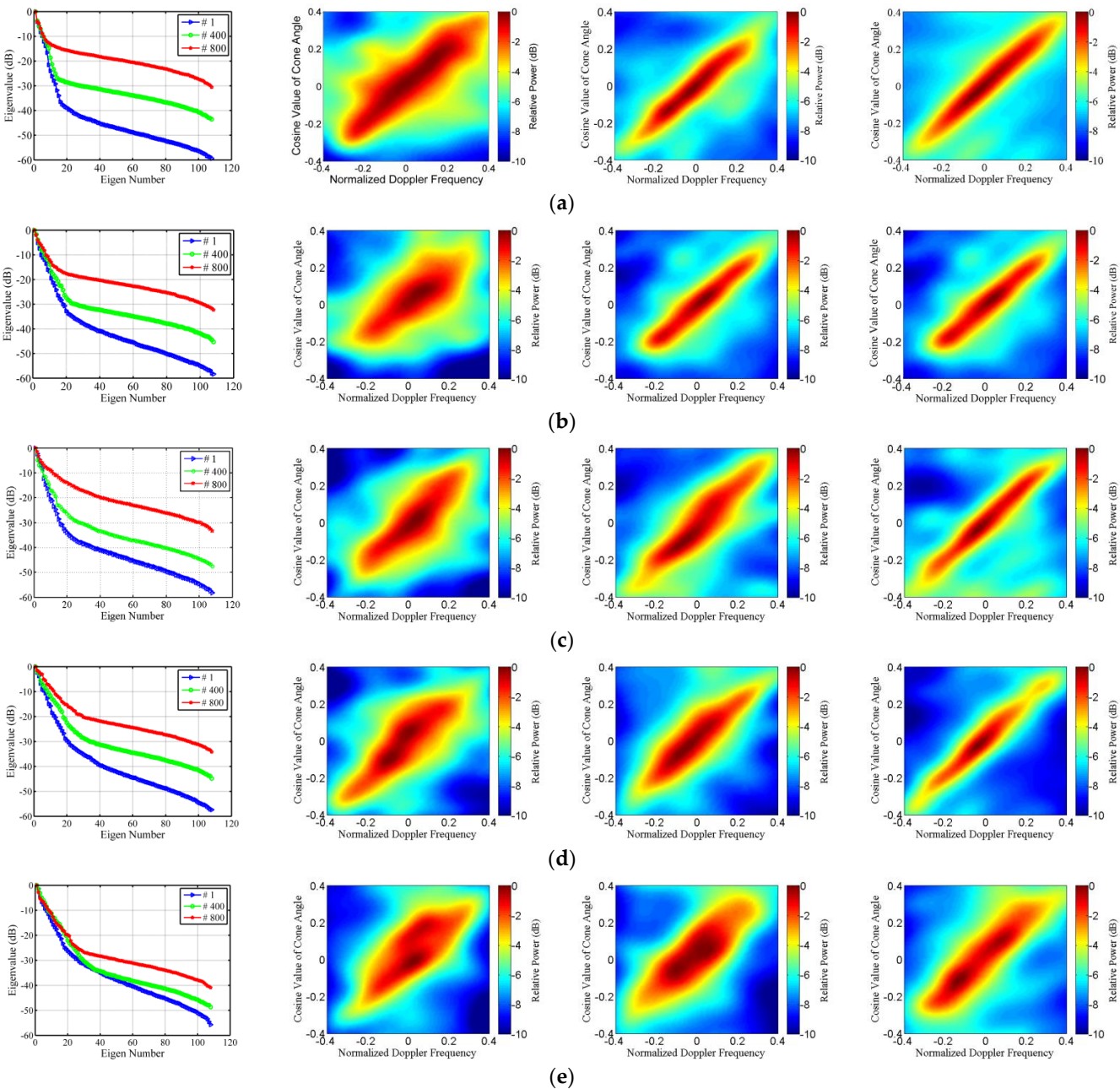

**Figure 10.** Eigenvalue spectrum and space-time power spectrum corresponding to different range gates (#1; #400; #800) under each sea state: (**a**) sea state 1; (**b**) sea state 2; (**c**) sea state 3; (**d**) sea state 4; and (**e**) sea state 5.

## 4. Discussion

### 4.1. The Space-Time Characteristics in Different Range Gates with the Same Sea State

Comparing the range of eigenvalues and the spread width of space-time power spectrum corresponding to each sea state in Figure 10, two obvious results can be drawn: (1) the range of eigenvalues decreases with the increase in the number of the range gate, especially under the sea states 1–4; (2) the spread width of the space-time power spectrum becomes narrower with the increase in the number of the range gate, especially under sea states 3–5; however, there is no obvious difference of the spread width of the clutter ridge between the range gate number 400 and 800.

From the space-time signal model of sea clutter in Section 2.2.2, it can be seen that the backscattering coefficient of sea clutter and the speed of the sea surface are the main vari-

ables that affect the space-time characteristics of sea clutter. In order to analyze the above results that come from the measured sea clutter data, the influence on the characteristics of space-time which caused by the backscattering coefficient of sea clutter and the speed of the sea surface are analyzed separately in the following.

For the speed of the sea surface which belongs to the different range gates in the same set of the clutter data, assume that the speed of the sea surface corresponding to each range gate which belongs to the main lobe area obeys the same distribution. This assumption is valid under the same sea state. According to the reference [21], there is a strong correlation between the bandwidth of the Doppler spectrum corresponding to the different range gates and the echo power of the sea clutter. However, if the operation as Section 2.3 is performed on the clutter data which used for Doppler spectrum estimation, the power attenuation caused by the range, elevation beam gain and the patch area are all compensated. Therefore, the difference of the Doppler spectrum corresponding to each range gate is only caused by the backscattering coefficient of sea clutter under different grazing angles.

In order to measure the temporal variation of the Doppler spectrum, we measured the Doppler centroid of the clutter spectrum $f_c$ and the root mean square (rms) bandwidth $B_w$, defined, respectively, as Equations (27) and (28).

$$f_c = \frac{\int_{-\infty}^{+\infty} f S(f) df}{\int_{-\infty}^{+\infty} S(f) df} \tag{27}$$

$$B_w = \sqrt{\frac{\int_{-\infty}^{+\infty} (f - f_c)^2 S(f) df}{\int_{-\infty}^{+\infty} S(f) df}} \tag{28}$$

where $f$ is the frequency and $S(f)$ is the clutter power spectral density (PSD). For example, we calculate the Doppler centroid of the clutter spectrum and the root mean square (rms) bandwidth under the sea state 3, which is shown in Figure 11. It can be seen from Figure 11 that the Doppler centroid and the rms bandwidth fluctuate above and below the mean value with the change of the range gate. The sea surface is roughness and time-varying, meanwhile, the power of clutter data fluctuates with the wind direction and wind speed. Therefore, it can be considered that the Doppler centroid and rms bandwidth corresponding to the Doppler spectrum of each range gate fluctuate with the number of the range gate, which is the manifestation of time-varying characteristics. Moreover, it can be concluded that the time decorrelation of each range gate which caused by the same sea state is nearly same under the condition of data compensation operation. So, the main influencing factor causing the variation of the eigenvalue range and the variation of the spread width of the space-time power spectrum is the large difference in the backscattering coefficient of sea clutter corresponding to the different range gates.

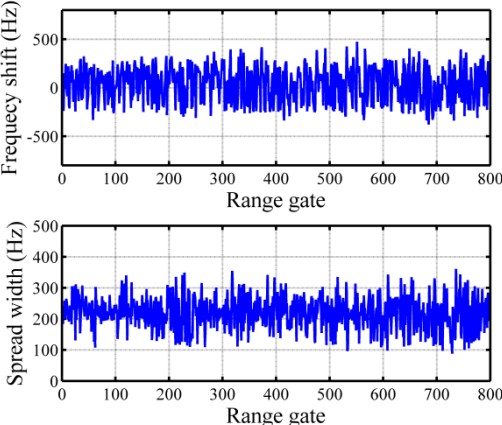

**Figure 11.** The Doppler centroid and root mean square bandwidth of the Doppler spectrum.

For the backscattering coefficient of sea clutter which belongs to the different range gates in the same set of the clutter data, there exists a large difference among the different range gates in the condition of the low grazing angle which according to the scattering coefficient model of sea clutter [24,25]. According to Equation (14), there is also a certain difference in the backscattering coefficient of sea clutter corresponding to each azimuth angle. Therefore, the space-time characteristics belonging to the sub-patch in azimuth is the smallest unit in which the characteristics of the sea clutter space time can be studied.

Moreover, the scattering coefficient of sea clutter varies with the pulse due to the motion of the sea surface. Therefore, the echo of the sub-patch in azimuth along the pulse $\mathbf{x}^t_{\theta,\varphi}$ is expressed as

$$\mathbf{x}^t_{\theta,\varphi} = (\sigma_m)^M_{m=1} \odot \boldsymbol{s}_t = (\sigma_m)^M_{m=1} \odot \boldsymbol{s}_{to} \odot \boldsymbol{s}_{tc} \tag{29}$$

where $(\sigma_m)^M_{m=1}$ is the backscattering coefficient of sea clutter under different pulses, and $\boldsymbol{s}_t$, $\boldsymbol{s}_{to}$, $\boldsymbol{s}_{tc}$ as shown in Equation (16). Both the scattering coefficient $\sigma_m$ and the steering vector $\boldsymbol{s}_{tc}$ result in decoupling between space and time and a large extension of the space-time spectrum of sea clutter.

Then, the temporal covariance matrix of the sub-patch in azimuth is written as

$$\begin{aligned} \mathbf{R}^t_{\theta,\varphi} &= \mathbf{x}^t_{\theta,\varphi} \mathbf{x}^{t\,H}_{\theta,\varphi} \\ &= \left(\boldsymbol{s}_{to}\boldsymbol{s}_{to}{}^H\right) \odot \left(\boldsymbol{s}_{tc}\boldsymbol{s}_{tc}{}^H\right) \odot \left((\sigma_m)^M_{m=1} \cdot \left((\sigma_m)^M_{m=1}\right)^H\right) \\ &= \left(\boldsymbol{s}_{to}\boldsymbol{s}_{to}{}^H\right) \odot \left(\boldsymbol{s}_{tc}\boldsymbol{s}_{tc}{}^H\right) \odot \mathbf{A}_t \end{aligned} \tag{30}$$

where $\mathbf{A}_t$ is the decorrelation matrix of the scattering coefficient. The spatial covariance matrix is assumed as $\mathbf{R}_s$, and the space-time covariance matrix of sea clutter is written as

$$\mathbf{R}_{s-t}(\theta, \varphi) = \mathbf{R}_t(\theta, \varphi) \otimes \mathbf{R}_s(\theta, \varphi) \tag{31}$$

From Equation (31), we can notice that the decoupling of the space-time covariance matrix is caused by the space decorrelation of the backscattering coefficient of sea clutter and the time decorrelation of the intrinsic motion of the sea clutter. However, the space-time signal is the sum of the space-time echo belonging to all the sub-patches in azimuth, which will lead to a certain difference between the space-time covariance matrix corresponding to the single range gate and that corresponding to the sub-patch. Therefore, the above theoretical analysis can only explain the spread width of the space-time spectrum and range of eigenvalues corresponding to the different range gates from the theoretical level, which can only give the qualitative conclusions and cannot give the quantitative conclusions. In order to solve this problem, the correlation analysis of the scattering coefficient of sea surface with the spread width of the space-time spectrum and the range of eigenvalues are carried out in the following section, based on the measured data.

Before conducting the cross-correlation analysis, there are several important concepts that need to be introduced. First, we define the equivalent backscattering coefficient of sea clutter of the range gate which can be calculated by the backscattering coefficient of sea clutter to system loss ratio, and the ratio can be obtained from the amplitude of sea clutter with the radar equation. For example, the equivalent scattering coefficient corresponding to each range gate can be obtained with the parameters in Table 3 and radar Equation (13), which are shown as the first figure in Figure 12a. Second, we define the 3 dB spread width which is the ratio of the power spectrum maximum value down by 3 dB to the power spectrum maximum value occupying the entire power spectrum plane. The 3 dB spread width under the sea state 3 corresponding to each range gate is shown as the second figure in Figure 12a. Third, we calculate the range of the eigenvalue corresponding to each range gate, which is shown as the third figure in Figure 12a. For the above three curves, the correlation between the equivalent backscattering coefficient of sea clutter and the 3 dB spread width and the correlation between the equivalent backscattering coefficient

of sea clutter and the range of the eigenvalues are analyzed with the cross-correlation function [29], which is shown in Figure 12b.

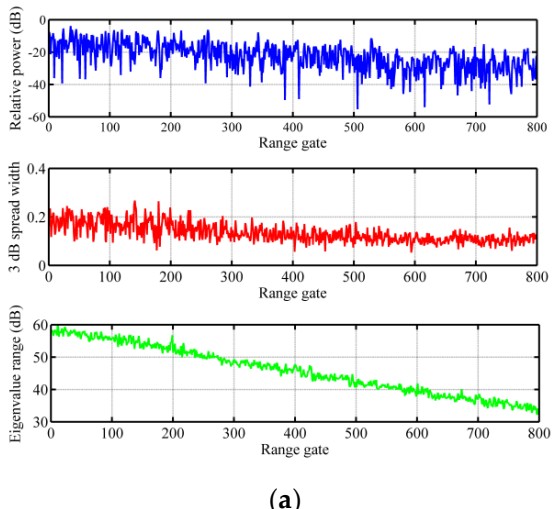 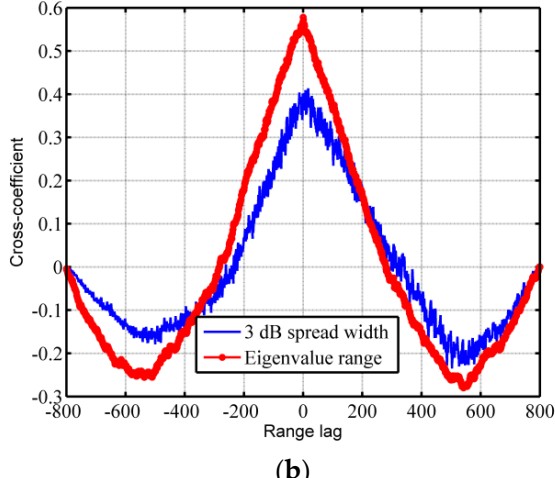

(**a**)　　　　　　　　　　　　　　　　　(**b**)

**Figure 12.** Cross-correlation analysis: (**a**) equivalent backscattering coefficient of sea clutter, 3 dB spread width and the range of the eigenvalue; (**b**) cross-covariance between equivalent backscattering coefficient of sea clutter and 3 dB spread width and between equivalent backscattering coefficient of sea clutter and the range of eigenvalue.

From the above cross-correlation coefficient curve in Figure 12b, there are two conclusions can be drawn in the following: (1) When the range delay is zero, the cross-correlation coefficient between the equivalent backscattering coefficient of sea clutter and the 3 dB spread width of the power spectrum is only 0.42, while the cross-correlation coefficient between the equivalent backscattering coefficient of sea clutter and the range of the eigenvalues is 0.59. It means that the equivalent backscattering coefficient of sea clutter mainly affects the range of the eigenvalues and also affects the spread width of the clutter ridge, where the effect is slightly smaller than the former. (2) When the range delay is not zero, the cross-correlation coefficient between the equivalent backscattering coefficient of sea clutter and the 3 dB spread width of the power spectrum decreases rapidly with the increase in the range delay number; however, after a certain range delay, there exists a small fluctuation in the correlation coefficient.

### 4.2. The Space-Time Characteristics in Different Sea States with the Same CNR

Comparing the spread width of the power spectrum and the eigenvalue number of each range gate under five sea states in Figure 10, two obvious results can be drawn: (1) for the eigenvalue number of the clutter, the change in the number of eigenvalues is proportional to the change in sea states under the same threshold; however, this phenomenon only exists in range gate 1, as shown in Figure 13a; (2) for the spread width of the space-time power spectrum, there is no obvious spectral broadening corresponding to the range gate 1; however, there are obvious changes in the spread width of the power spectrum corresponding to the range gate 400 and 800 with the change of the sea states, as shown in Figure 13b,c. The comparative results above differ to the conclusion shown in [17], that the clutter eigenvalue is bigger under the high sea state than that under the low sea state, while the clutter spectrum is wider under the high sea state than that under the low sea state.

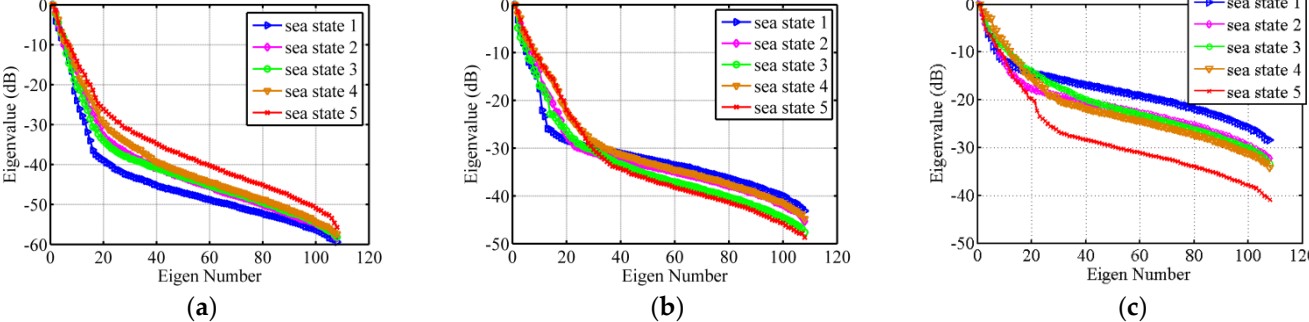

**Figure 13.** Comparison of the eigenvalue spectrum corresponding to different range gates: (**a**) #1 range gate; (**b**) #400 range gate; (**c**) #800 range gate.

Why does the above result appear? It is known that the clutter amplitude will increase with the increase in the sea states, which is caused by the increase in the backscattering coefficient of sea clutter with the increase in the sea states. Therefore, there are different CNRs of the same range gate under the different sea states. The above comparison of the space-time characteristics corresponding to the same range gate is actually a comparison of that under different CNR. From Section 4.2, it can be seen that the space-time characteristics of the sea clutter under different CNRs are quite different. Obviously, the reason for the above result is that the space-time covariance matrix of sea clutter is affected by different CNR and different speeds of the sea surface motion, which can be seen in Equation (30). Before analyzing the space-time characteristics of sea clutter at different speeds of the sea surface motion, it is necessary to select the range gate with the same CNR according to the echo amplitude under the various sea states. The curve of the CNR is shown in Figure 14.

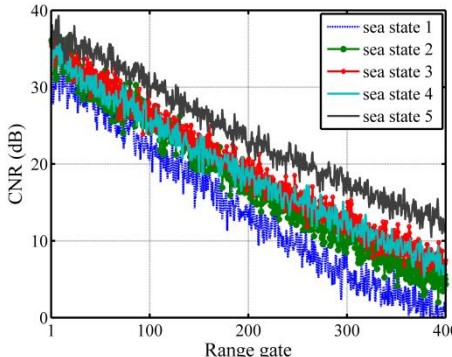

**Figure 14.** The CNR corresponding to different range gates.

Without loss of generality, we select the range segment with less CNR variation corresponding to the five sea states, and find the range gate number belonging to each sea states. The range gate numbers used in Figure 14 are 110, 130, 190, 180, 240, corresponding to the sea state 1, 2, 3, 4, 5, respectively. That is, the corresponding CNR under the above five sea states is 20 dB. Under the condition of the same CNR, the eigenvalue spectrum and space-time power spectrum corresponding to five sea states are estimated with the method introduced in Section 2.3, as shown in Figure 15.

It is assumed that the equivalent backscattering coefficient of sea clutter corresponding to five sea states is nearly the same when the CNR is same. Figure 15 shows that the eigenvalue number of the sea clutter is bigger under the higher sea state than that under the lower sea state, when the eigenvalue number of sea clutter is determined by the same threshold. Meanwhile, Figure 15 shows that the spread width of the power spectrum is wider under the higher sea state than that under the lower sea state. This is mainly caused by the speed of sea surface motion increasing with the sea states. Therefore, the faster speed

of the sea surface motion in higher sea state leads to more decoupling of spatial frequency and Doppler, which is consistent with the theoretical analysis in Section 2.2.2.

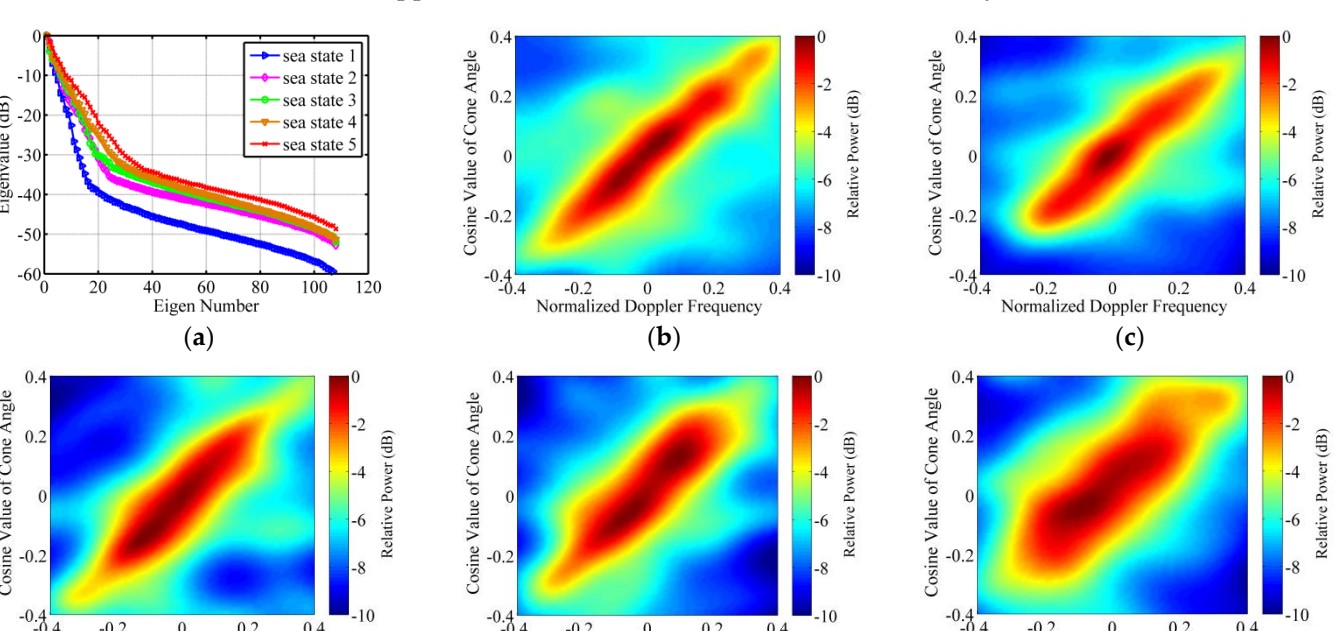

**Figure 15.** Comparison of the space-time characteristics in different sea states with the same CNR: (**a**) eigenvalue spectrum under 5 sea states; (**b**) power spectrum under sea state 1; (**c**) power spectrum under sea state 2; (**d**) power spectrum under sea state 3; (**e**) power spectrum under sea state 4; and (**f**) power spectrum under sea state 5.

Sections 4.1 and 4.2, respectively, analyze the decorrelation effects of the backscattering coefficient of sea clutter and the speed of sea surface motion on the space-time characteristics of sea clutter. The comprehensive comparison shows that the decorrelation effect caused by the change of the sea surface scattering coefficient is greater than that caused by the speed of the sea surface motion.

## 5. Conclusions

In this paper, we introduce, in detail, the method and experimental equipment for acquiring sea clutter data with space-time coupling characteristics based on an L-band shore-based multichannel radar. The sea clutter with space-time characteristics can be measured by a transmit array that moves periodically with the PRI, and the space-time matrix can be accurately estimated by using the data composed of the single range gate and multiple CPIs. The processing results of the simulated and measured data show that the sea clutter data with the space-time characteristics can be validly acquired by this measurement method. The space-time characteristics corresponding to each range gate can be easily obtained by using this measurement mode, which can provide a research basis for the study of sea clutter inhomogeneity.

For the space-time characteristics in different range gates under same sea state, comparing the cross correlation results between the equivalent scattering coefficient of sea clutter and the range of eigenvalues, and comparing that between the equivalent backscattering coefficient of sea clutter and 3 dB spread width of clutter ridge, shows that the main influencing factor causing the variation of eigenvalue range and spread width of space-time power spectrum is the large difference of the scattering coefficient of sea clutter existing in different range gates. The reason why the backscattering coefficient of sea clutter varies greatly in different range gate is that the backscattering coefficient of sea clutter varies greatly in its low grazing angle. By comparing the space-time characteristics in different

sea states with the same CNR, it can be concluded that the faster speed of the sea surface motion in higher sea state leads to more decoupling of spatial frequency and Doppler. The comprehensive comparison of the effect of the scattering coefficient of sea clutter and the speed of the sea surface motion shows that the decorrelation effect caused by the change of the backscattering coefficient of sea clutter is greater than that caused by the speed of the sea surface motion when the radar works in a low grazing angle.

Furthermore, the multichannel sea clutter measurement method proposed in this paper can be used for measuring sea clutter in different ocean parameters, which can provide multiple sea clutter data for analyzing inhomogeneity of sea clutter space-time characteristics. Moreover, the results of sea clutter space-time characteristics can also provide the reference for airborne microwave radar systems in order to detect vessels in the open sea working at low grazing angles.

**Author Contributions:** J.W. designed the space-time signal model and wrote the manuscript. F.L. supervised the preparation of the manuscript and coordinated revisions. Y.Z. designed the multi-channel sea clutter measurement system. J.Z. provided key guidance in this article. X.X. measured and processed data. All authors have read and agreed to the published version of the manuscript.

**Funding:** This research was funded by National Natural Science Foundation of China under Grant number U2006207.

**Institutional Review Board Statement:** Not applicable.

**Informed Consent Statement:** Not applicable.

**Data Availability Statement:** Not applicable.

**Conflicts of Interest:** The authors declare that they have no conflict of interest.

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
