# Peer review of "Multichannel Sea Clutter Measurement and Space-Time Characteristics Analysis with L-Band Shore-Based Radar"

_remotesensing, doi:10.3390/rs14215312_

Round 1

Reviewer 1 Report

The paper is well written, and contributes a method for multichannel sea clutter measurement based on L-band shore-based radar, which can exclude the influence of the moving platform and measure the pure sea clutter data with the space-time coupling characteristics. And the space-time characteristics of sea clutter are analyzed based on the measured data, and the results are helpful for studying the clutter suppression algorithms of the marine multichannel radar. This is an interesting and exciting manuscript that certainly merits publication. However, there are some minor problems in the following, which must be solved before publishing.  Please see the attachment.

Reviewer 2 Report

(1) The sea clutter measurement system of the emulated airborne movement is implemented based on the shore-based multichannel radar.  And IDPCA technology plays a crucial role in this process. However, there is little description about how to use IDPCA technology in shore-based radar in this paper. Please supplement it appropriately.

(2) In equation 14 on page 9, which is the scattering coefficient of the clutter and which is the sea surface radial velocity?

(3) Is the legend of the transmitting antenna and receiving antenna in Figure 1 on page 4 correct? Are the arrows opposite?

(4) Figure 10 on page 20 is not clear. It is recommended to remove the circles and triangles, which can make the frequency variation and range to be seen more clearly.

(5) Where is Figure 15?  Which is mentioned in line 714, but it is not in the manuscript. 
